# The MUDENG Augmentation: A Genesis in Anti-Cancer Therapy?

**DOI:** 10.3390/ijms21155583

**Published:** 2020-08-04

**Authors:** Manikandan Muthu, Sechul Chun, Judy Gopal, Gyun-Seok Park, Arti Nile, Jisoo Shin, Juhyun Shin, Tae-Hyoung Kim, Jae-Wook Oh

**Affiliations:** 1Department of Environmental Health Sciences, Konkuk University, Seoul 143-701, Korea; bhagatmani@gmail.com (M.M.); scchun@konkuk.ac.kr (S.C.); jejudy777@gmail.com (J.G.); 2Department of Bioresources and Food Science, Konkuk University, Seoul 143-701, Korea; bhagatmani@yahoo.co.in (G.-S.P.); nileshivraj@yahoo.com (A.N.); 3Department of Stem Cell and Regenerative Biotechnology, Konkuk University, Seoul 143-701, Korea; Oh@gmail.com (J.S.); judy.je.gopal@gmail.com (J.S.); 4Department of Biochemistry, Chosun University School of Medicine, 309 Pilmoondaero, Dong-gu, Gwangju 501-759, Korea; thkim65@chosun.ac.kr

**Keywords:** cancer, MUDENG, therapy, TRAIL, AP5M1, BAX

## Abstract

Despite multitudes of reports on cancer remedies available, we are far from being able to declare that we have arrived at that defining anti-cancer therapy. In recent decades, researchers have been looking into the possibility of enhancing cell death-related signaling pathways in cancer cells using pro-apoptotic proteins. Tumor necrosis factor (TNF)-related apoptosis-inducing ligand (TRAIL) and Mu-2/AP1M2 domain containing, death-inducing (MUDENG, MuD) have been established for their ability to bring about cell death specifically in cancer cells. Targeted cell death is a very attractive term when it comes to cancer, since most therapies also affect normal cells. In this direction TRAIL has made noteworthy progress. This review briefly sums up what has been done using TRAIL in cancer therapeutics. The importance of MuD and what has been achieved thus far through MuD and the need to widen and concentrate on applicational aspects of MuD has been highlighted. This has been suggested as the future perspective of MuD towards prospective progress in cancer research.

## 1. Introduction

Genes involved in balancing cell proliferation and cell death may undergo mutation, disrupting tissue homeostasis culminating in rapid multiplication of cancerous cells [1,2,3,4]. Such cells that show uncontrollable cell division result in tumors. Cancer is currently one of the leading causes for premature mortality. The criteria to choose the best treatment totally depends upon the type of cancer (metastatic or benign), location (lymphatic or sarcoma), and exaggeration of the cancerous cell as well as the immune system of an individual. Although improvements in detection methods, clinical intervention, and increased public awareness of risk factors are in practice, cancer is on the rise [5,6,7]. The predominant treatment since 1800s has been surgery to remove the bulk of the tumor followed by radiotherapy and/or chemotherapy to destroy remnant cancerous cells. Treatment has undergone a slow evolution and four mainstream modes of treatment have been recognized. The first being surgery. The second treatment module is radiotherapy, established at the end of the 19th century, which utilizes X-rays and/or G-rays to block essential biochemical processes resulting in cell death [8]. The third development, chemotherapy, through introduction of chemotherapy agents (analogues of nitrogen mustard gas). Chemotherapy agents have since been used in combination to prevent the development of resistance. The fourth development is precision therapy or targeted therapy. This initialized with the discovery of a small-molecule kinase inhibitor specifically targeted to the mutant BCR-ABL protein present in the tumor cells of patients with chronic myeloid leukemia (CML). This concept has now become the ‘gold standard’ approach for discovering new cancer treatments. Over the past decade, immuno-oncology (IO) has emerged as a novel and important approach to cancer treatment through the stimulation of the body’s own immune system to kill cancer cells. Yet another strategy is also underway to enhance cell death-related signaling pathways in cancer cells using proapoptotic proteins [9,10,11,12,13,14,15].

In the mid-1990s, a new member of the tumor necrosis factor (TNF) family was discovered and named TNF-related apoptosis-inducing ligand (TRAIL) [16,17]. TRAIL was shown to possess the ability to induce apoptosis in a wide range of human cancer cell lines without significant cytotoxicity towards normal cells [18,19,20]. The pursuit for new therapies continues and various gene related therapies, hormone therapies, and their achievements have been reported [21]. With nanotechnology impacting every area of science and development and innovative medicine, why would it spare cancer alone? The effectiveness of nanotherapy for cancer is dependent on precise medicine, precise dose, accurate target, precise time of exposure, patient safety for precise therapeutic output.

Nano-drug delivery systems have successfully delivered a payload of chemotherapeutic agent to the specific site [22]. Nano-delivery systems could aid in precise and effective drug dosing with reduced side effects [23]. Nano-sized drugs have better invasive ability and effectively penetrate the targeted cancerous tissue through the vascularized system, using its enhanced permeability and retention (EPR) property [24]. This EPR effect of nanodrugs improves the delivery of the drug molecule, enabling treatment at reduced concentrations [25,26]. A synergistic nanotheranostic formulation combined with other cancer therapies such as photothermal therapy is also reported for enhanced diagnosis and elimination of tumors [27]. Carrier-assistant drug delivery systems (DDSs) have made effective progress in cancer diagnosis and treatment [28,29,30,31,32,33,34]. Thus, nanotherapy is definitely a new hope for cancer patients but the disadvantages include higher treatment cost, risks of chronic toxicity, and limited clinical testing data. Currently, researchers are concentrating on overcoming these disadvantages through fabrication of non-toxic, cost-effective, engineered nanomaterials [35,36].

This review first describes the potential of using TRAIL-R agonists for cancer therapy. A more conceptually newer topic, Mu-2/AP1M2 domain containing, death-inducing (MUDENG, MuD), and its valuable achievements have been discussed elaborately. This review further presents the deliverables from MuD in cancer research. This review envisions a promising future for MuD in cancer therapy. The research areas within the capacity of the MuD theme worth refining are also speculated.

## 2. Trail of TRAIL in Cancer Therapeutics

In the recent decades, adequately equipped chemotherapy and combinatorial treatment programs have diminished death rates from cancer. The strategy for proving the treatment efficacy has now entered a new era; applying the rationale behind the mechanism(s) of action, interactions, and pharmacology and investing this knowledge whilst planning combat strategies. The advent of selective therapeutic approaches holds promises towards even more successful outcomes. TRAIL system is one such highly evolved approach showing encouraging levels of efficacy. Ever since its identification in 1995 [16,17], TRAIL has evoked growing interest in oncology owing to its reported ability to selectively target cancer cells. There are other entities belonging to the TNF superfamily, but contrary to them all, TRAIL administration in vivo has proven to be harmless. The relative absence of toxic side effects of this naturally occurring cytokine, in addition to its antitumor properties has led to its consideration as a resourceful hope towards anti-cancer therapeutics. TRAIL has now been recognized to selectively eradicate cancer cells by activating a signaling pathway that is used by the innate immune system (apoptosis) [37]. Figure 1 gives an overview of the currently available cancer therapies and the multifaceted approaches of TRAIL in cancer therapy.

Apoptosis is nature’s way for harmoniously maintaining tissue homeostasis by systematically eliminating harmful cells from the body [38]. There are two pathways through which apoptosis may occur, namely the extrinsic and intrinsic pathways. The extrinsic pathway is mediated by death receptors belonging to the TNF receptor superfamily such as TRAIL-R1/R2 [39]. The intrinsic pathway alternatively also known as the mitochondrial pathway, is triggered in response to cellular stress as well as DNA damage. This then triggers the activation of p53 and the release of pro-apoptotic factors from the mitochondria [40]. TRAIL-induced apoptosis is primarily mediated by the extrinsic pathway, but when a cell undergoes added stress it can be enhanced by the intrinsic pathway, resulting in expedited apoptosis [41]. It is documented that cross-talk occurs at various points involving the extrinsic and intrinsic apoptotic signaling pathways, orchestrating a complex and intricately balanced flow [42]. One such common node is caspase-8, that cleaves differential substrates in both pathways [43]. To maintain control of the apoptotic machinery, both these pathways are regulated by pro-apoptotic and anti-apoptotic modulators. It is noteworthy that TRAIL signaling does not necessarily end in apoptosis of cancer cells. Studies have shown that TRAIL may induce pro-survival response via signaling factors that include nuclear factor (NF)-κB, mitogen-activated protein kinase (MAPK), and Akt (protein kinase B) [44]. The pro-apoptotic and pro-survival signals vie with each other determining the survival outcome.

Numerous recombinant versions of human TRAIL have been developed to augment its tumor-killing potential. Naturally occurring untagged soluble human TRAIL has a short serum half-life of approximately 30 min in non-human primates [45]. Owing to its small size, TRAIL is rapidly cleared from the body via the kidneys [45,46]. This rises the demand for repeated administration or towards improved delivery methods to maintain effective levels within the human body [47]. One approach has been to facilitate its oligomerization by addition of peptide ‘tags’, this increases the total size of the oligomer. Whereby, it has a two-fold effect, by reducing its in vivo clearance and also by decreasing the proportion of inactive aggregates. Addition of a FLAG tag (FLAG-TRAIL) [48], leucine zipper (LZ-TRAIL) [19]; iz-TRAIL [49,50], or a tanascin-C (TNC) oligomerization domain (TNC-TRAIL) [51] are few examples to this strategy. These modifications are also known to enhance TRAIL activity, besides enhancing the stability. Alternatively, the pharmacokinetic profile can be enhanced by covalently linking it to a molecule with more favorable properties [52]. Few researchers have added human serum albumin [23] (which has a superior plasma half-life compared to that of TRAIL) to the N terminus of TRAIL. This modification is reported to significantly improve its circulating half-life in vivo whilst maintaining its antitumor activity [53]. Polyethylene glycol (PEG) has also been attempted to tailor-make PEGylated TRAIL derivatives. This exhibited protracted antitumor activity compared to the untagged TRAIL [54,55]. Kim et al. have used a nanocomplex system rather than a chemical modification, leading to enhancement of its long term delivery properties [56]. Kim et al. have demonstrated using an in vivo xenograft tumor model, that TRAIL-loaded microspheres inhibited tumor growth and displayed sustained TRAIL release for a period of >10 days [56]. These approaches have helped to improve the therapeutic efficacy of systemically delivered TRAIL. Table 1 summarizes the achievements of TRAIL in anti-cancer therapeutics.

As do all biological systems, tumor cells too have exceptions and mixed response to TRAIL-mediated killing. Researchers have reported that untagged TRAIL displayed some amount of cytostatic or cytotoxic effects on multiple cancer cell lines, but 20% were refractory to its action [20]. Furthermore, it is also known that cancer cells acquire TRAIL resistance during the evolution of the tumor [75]. Intrinsic and acquired resistance to TRAIL is something that poses a huge problem in establishing clinically efficacious TRAIL therapies. This resistance can show up anywhere between binding and cleavage sites of effector caspases, along the TRAIL-signaling pathway [41,76]. Many tumor cell lines are TRAIL resistant [19] and this resistance is reported to vary within primary human tumor cells. Since TRAIL signaling is an important part of the host mechanism to suppress tumor formation and metastasis [20,77], advanced cancers often evolve TRAIL resistance. Considering the proportion of cancer cells with some degree of intrinsic or acquired resistance towards TRAIL, synergistic approaches have also been considered. The cancer resistance to TRAIL has led to combinational therapy to restore apoptosis sensitivity [78]. This has been accomplished through chemotherapy drugs or epigenetic modulators or autophagy inhibitors. These are able to sensitize TRAIL resistant cancer cells towards TRAIL-induced apoptosis through various molecular mechanisms. Other mechanisms of TRAIL resistance include aberrant protein synthesis, protein misfolding, ubiquitin regulated death receptor expression, metabolic pathways, and metastasis [79]. Piras et al. [80] theoretically demonstrated that by targeting a molecule at the survival and apoptosis pathway junction, TRAIL-resistance can be overcome. It suppresses JNK and, at the same time, enhances caspases activities. Various classes of drugs have been used in combination with TRAIL [81].

From an apoptotic viewpoint, TRAIL initiates cell death in a p53-independent manner [41] and chemotherapy or radiation are often p53-dependent, thus a combination of the two treatments becomes fruitful [81]. Recombinant soluble TRAIL modification is reported to exhibit strong tumoricidal activity against glioblastoma multiforme (GBM) cells with nil to minimum toxicity against normal cells [66]. However, still, no single therapeutic agent including TRAIL has offered a perfect solution with 100% success rate [18,47]. Various combinatorial strategies, considering conventional or novel targeted therapies, have been considered for the synergistic enhancement of TRAIL activity [17,18]. Although a lot of reports have been published [76,79,82,83,84,85,86,87,88,89,90,91,92] with respect to TRAIL resistance, detailed resistance mechanisms have not been elucidated and it is still unknown whether different types of cancer undergo TRAIL resistance through similar or specific mechanisms.

## 3. Evolution of MUDENG

### 3.1. What Is MUDENG

Mu-2/AP1M2 domain containing, death-inducing (MUDENG), is alternatively known as MuD or putative HIV-1 infection-related protein. It is a 490 amino acid protein belonging to the adaptor complexes (APs) medium subunit family. MuD possesses an adaptin domain found in the Mu-2 subunit of APs related to clathrin-mediated endocytosis which is able to independently induce cell death. MuD is expressed in most tissues, including intestine and testis, encoded by a gene on the human chromosome 14q22.3. This gene has been evolutionarily conserved in some mammals, suggesting that it has a universal role in cell death. MuD is reported to exist as two alternative spliced isoforms and expressed in Jurkat T-cells, HepG2 cells, and HeLa cells tumor cell lines. MuD is a novel gene associated with Fas-mediated apoptosis [93]. Further investigations on MuD revealed a Mu (μ) homology domain (found in adapter proteins) that plays a major role in intracellular trafficking pathways and orchestrates cell death in cytotoxic T cells [94]. Hirst et al. [95] recently reported that C14orf108 is a component of adapter protein-5, that is involved in endosomal trafficking, eventually it is now confirmed that *C14orf108* and *MuD* are one and the same gene [96].

### 3.2. Mechanism of MuD Based Anti-Cancer Therapy

Apoptosis is a necessary process to maintain physiological balance by removing unnecessary cells in the body [38]. TRAIL-mediated apoptosis occurs by the extrinsic pathway (Figure 2) [97]. In TRAIL-medicated apoptotic signaling, various pro-apoptotic and anti-apoptotic proteins within cells are activated to suit the situation, maintaining the ideal balance on the path to apoptosis and survival. In these circumstances the cross-talk process leading to the activation of caspases-8/-9/-3 is critical. Based on our previous results, MuD appears to be operating within the mitochondria-mediated pathway. MuD resides mainly at the endoplasmic reticulum (ER) and the mitochondria (to a certain extent), regulates apoptotic signaling induced by the formation of TRAIL/TRAIL-R complex. Bid molecules are involved in apoptosis and caspase-3 may be essential for the regulation of MuD expression and function. In addition, the formation of 25-kDa Bcl-2 fragments by TRAIL stimulation, which converts Bcl-2 from its anti-apoptotic form to the truncated pro-apoptotic form, is inhibited by MuD. This affects the regulation of caspase activation and subsequent apoptosis [62,64].

MuD if overexpressed by transfection in GBM cell lines such as U251-MG cells, cells become more resistant to TRAIL stimulation, however if MuD is knocked-down through siRNA transfection in same cells, TRAIL resistance disappears (about 20% survival rate). As well, results from previous experiments confirmed that MuD was mostly present in ER and partly in mitochondria. In extrinsic TRAIL-medicated apoptotic signaling, MuD was confirmed to be involved in mitochondria-mediated apoptotic signaling. Specifically, it appears to have an ani-apoptotic function between Bid and Bcl-2 apoptotic factors upon TRAIL stimulation.

TRAIL and MuD never interact with each other during apoptotic signaling. Trimeric complex, TRAIL, is a ligand of TRAIL receptors (TRAIL-R) called DR4/5. However, the connective junction between the two (proved through experiments) was between Bid and Bcl-2 apoptotic factors in TRAIL-medicated apoptotic signaling on the extrinsic pathway. Based on the research results so far, the possible approach using target MuD for cancer treatment is to suppress the expression of MuD. This is orchestrated using a specific siRNA targeted MuD knock-down or knock-out (KO) of the targeted MuD using the CRISPR/Cas9 system (we are using this technique to generate MuD KO cells). This is one of combination methods with TRAIL or chemo-/radio-therapeutic agents for clinical approaches.

### 3.3. MuD Milestones in Therapeutics

We reviewed MuD and its deliverables in the arena of therapy, and compiled the available reports on MuD and present it here. The current up-to-date achievements of MuD were reviewed. Apoptotic cell death is mediated by many genes that may be the key players in the induction and propagation of apoptotic cell death that is set off by death stimuli. Lee et al. [94] initiated the pioneering study to determine if MuD is able to induce cell death. HeLa cells were transfected with MuD expression plasmid constructs using a microporator. Their results conclusively confirmed that the wild type MuD significantly induced cell death. They reported that ectopic expression of MuD induced cell death in both Jurkat T cells and HeLa cells. Their results pointed out to the fact that MuD was likely to play an important role in cell death in various cancer cell lines. Wagley et al. [98] reported the successful generation of a mouse monoclonal antibody (MAb) against the middle domain of human (h) MuD. This IgG_1_ subtype MAb, named M3H9, recognizes residues 244–326 in the middle domain of the MuD protein. Similarly, MuD proteins expressed in astroglioma cell lines and primary astrocytes were also detected by M3H9 MAb. M3H9 MAb was demonstrated to be able to detect the expression of the MuD protein in formalin-fixed, paraffin-embedded mouse ovary and uterus tissues. These results pointed out to the fact that MuD Mab M3H9 may be suitable candidates for a new biomarker of hereditary spastic paraplegia, cancer, and other related diseases. The MAbs were able to detect proteins approximately 54 kDa in size in the brain cell lysates, and could be used on MuD detection in normal and cancer cells of human origin. Later, it was reported that overexpression of MuD in cells induces cell death [94,99], supporting the theory that MuD may have an important role in cell death or survival. In addition, a recent study has reported that MuD may interact with KIAA0415/SPG48 and play an important role in intracellular trafficking pathways [95]. MuD and KIAA0415/SPG48 are the two subunits of AP-5 complexes.

The *MuD* gene was first identified during screening of novel genes associated with Fas-mediated apoptosis [93]. Fas (CD95/APO-1) and TRAIL (CD253, TNFSF10, APO2) receptor are members of a subset of the TNF receptor superfamily known as ‘death receptors’ [100]. To date, the overwhelming majority of studies on Fas and TRAIL-R have explored the role of these receptors as initiators of apoptosis. It is reported that although transformed cells frequently express Fas and TRAIL-R, most do not undergo apoptosis interacting with these cognate ligands. Significant effort has been devoted towards estimating the sensitivity of such cells to the pro-apoptotic effects of ‘death receptor’ stimulation. The expression of Fas and TRAIL receptors is said to be greatly elevated in many cancer types such as hepatocellular carcinoma, renal carcinoma, and ovarian cancer.

Shin et al. [101] demonstrated that MuD is cleaved by caspase-3 during TRAIL-induced cell death. Their report highlighted the significance of TRAIL as well as MuD. MuD, also known as the adaptor-related protein complex 5 subunit mu 1 (AP5M1), was originally identified to induce cell death in lymphoma cell lines. The mechanism responsible for MuD-mediated cell death was investigated and the series of changes it goes through during TRAIL-induced cell death was established. MuD was rapidly processed in response to TRAIL in Jurkat and BJAB cells concomitantly to caspase activation. Caspase-3-mediated MuD cleavage was verified by an in vitro cleavage assay. Caspase cleavage sites (D276 and D290) were found to be located on the adaptin domain of MuD, and once cleaved, the killing ability of MuD decreased. These results suggest that the adaptin domain plays a key role in MuD-mediated cell death. This study shows that caspase-3 cleaves MuD during TRAIL-induced cell death, resulting in reduced MuD-induced cytotoxicity. The results emphasized that the adaptin domain of MuD may play a key role in its cytotoxicity, because caspase-3 cleavage sites are located in this domain.

Recently, Won et al. [102] reported the correlation between AP5M1 otherwise known as MuD and B cell lymphoma (Bcl)-2 Associated X [103], Apoptosis Regulator [104]. They demonstrated the potency of AP5M1 to induce apoptosis in cervical cancer cells. AP5M1 upregulated the level of BAX protein, a key pro-apoptotic Bcl-2 family member regulating the mitochondrial apoptotic cell death pathway. Moreover, AP5M1 completely lost its apoptotic activity in BAX-knockout or -knockdown cells, indicative of its functional dependence on BAX. Comparative analysis of cervical tissues from patients with cervical carcinoma and non-cancer controls revealed a prominent downregulation in AP5M1 expression with a concomitant downregulation in BAX expression; AP5M1 and BAX mRNA expression levels in cervical tissues exhibited a strong positive correlation. This report confirmed the association between AP5M1 and malignancy.

Oh et al., working on astrocytes (major glial cell type in the central nervous system which are sources of brain tumors), examined the MuD expression and function in human astroglioma cells elaborately. Their reports indicated that the stimulation of U251-MG cells with TRAIL resulted in a 40% decrease in cell viability and a 33% decrease in MuD protein levels. Decrease of MuD leads to reduced TRAIL stimulation in cells, leading to reduced cell viability. In addition, MuD depletion increased the susceptibility of the cells to TRAIL by enhancing the cleavage of caspase-3/-9 and BH3-interacting domain death agonist (Bid). The TRAIL-mediated decrease in cell viability in MuD-depleted cells was connected to Bid depletion, confirming the MuD site as the site of apoptotic signaling at the Bid and Bcl-2 junction. MuD localizes predominantly in the ER and partly in the mitochondria. The concentration of MuD is reduced on TRAIL stimulation, through caspase-3-mediated MuD cleavage. Thus, Oh et al. concluded that MuD not only possesses an anti-apoptotic function but may also constitute an important target for the design of ideal candidates for combinatorial treatment strategies for glioma cells [96].

In yet another interesting study Oh et al. have established that MuD possesses anti-apoptotic functions and is involved in silver nanoparticle (AgNP)-induced astroglioma hormesis [105]. Their investigations on whether silver nanoparticles (AgNPs) induce hormesis in astroglioma cells, towards probing the possible involvement of MuD in AgNP-induced hormesis, has thrown more light on the functionality of MuD. AgNPs exhibited cytotoxic effects on cell proliferation in a dose-dependent manner. Increased MuD expression was observed during AgNP-induced astroglioma hormesis. Studies using *MuD*-knockout cells and *MuD* siRNA transfection showed that MuD might influence cell viability upon AgNP treatment. In addition, apoptotic cell population and production of reactive oxygen species in the absence of *MuD* were significantly increased. The phosphorylation of two mitogen-activated protein kinases, p38 and extracellular signal-regulated kinase (ERK), but not c-Jun N-terminal kinases (JNK), was observed upon AgNP stimulation. In summary, AgNPs at low doses induced hormesis of human astroglioma cells, and MuD and p38/ERK mediators are involved in AgNP-induced astroglioma hormesis, resulting in beneficial effects from the cellula r point of view.

Oh et al. [106] have also reported the anti-astroglioma effect of *Graviola* extracts from a MuD perspective: extracts from *Graviola* leaves, fruits, and seeds against human astroglioma cells. Graviola is established for its anti-cancer effect [107,108,109,110,111,112,113,114,115,116,117,118,119,120,121,122,123,124,125]. Further studies in this direction could help to shed more light on the exact mechanism behind this MuD perspective of *Graviola* extracts. The *Graviola* extracts were observed to exhibit significant cytotoxic effects on cell proliferation in a dose-dependent manner and altered MuD expression patterns.

Lv et al. [126] comprehensively surveyed specifically expressed genes (SEGs) using the SEGtool, based on the big data of gene expression from The Cancer Genome Atlas (TCGA) and the Genotype–Tissue Expression (GTEx) projects. In 15 solid tumors, 233 cancer-specific SEGs (cSEGs) were identified, these were specifically expressed in one cancer type and held high potential for being established as diagnostic biomarkers. Three cSEGs (OGDH, MUDENG, and ACO2) were confirmed to have a sample frequency > 80% in kidney cancer, suggesting their high sensitivity. Lv et al. thus validated the presence of MuD in kidney cancer and its potential to be treated as a biomarker.

## 4. Future of MuD: Challenges and Scope

Reviewing what has been achieved and the knowledge gained through MuD towards cancer, it is unusual that not much popularity or attention has been given to MUDENG. As we have presented in the review, except for limited research groups, nobody appears to be probing into the furtherance of this knowledge. Except for few cancer types, MuD has also not been applied to or worked out in the various forms of cancers cell lines and most of the MuD work appears to be surrounding astroglioma cells. This review hopes to evoke an interest into this area, and active application of this knowledge for cancer therapy.

Efforts are being made to study the functions of MuD in depth. It is predicted that activated MuD will not work independently in the apoptosis signaling process and will perform its function in cooperation with already known or unknown partner molecule(s). Recently, efforts to find a partner molecule using the TurboID system have been reported [127]. The combination of these TurboID systems to find MuD’s partner membrane will help reveal new functions of MuD involved in the cell death process. This revelation is expected to help MuD and its credentials to be put to best use.

There is a lot that can be done with MuD, and to MuD: in vivo studies, clinical trials, and troubleshooting modifications such as those done in TRAIL. Practical implications of MuD are far from being underway. There is definitely room for more and promise for more too. Fundamental research has somehow led to the establishment of the functionality of MuD. It is now time to evolve to the next phase, where application of MuD in real time clinical trials for improvisation as per specific requisites should ensue. TRAIL has been reported successfully for cancer therapy in combination with nano, radiation, photothermal systems, MuD has not been attempted in such synergistic applications. We propose the possibility of employing CRIPSR-based KO or siRNA for effective targeted cancer therapy. Being a conceptually newer form than TRAIL, MUDENG deserves and demands scrutiny.

## Figures and Tables

**Figure 1 ijms-21-05583-f001:**
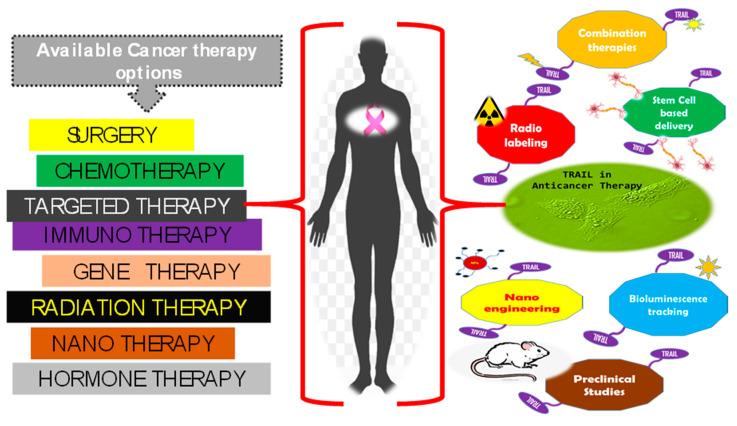
Overview of cancer therapies available and TNF-related apoptosis-inducing ligand (TRAIL) mediated anti-cancer options.

**Figure 2 ijms-21-05583-f002:**
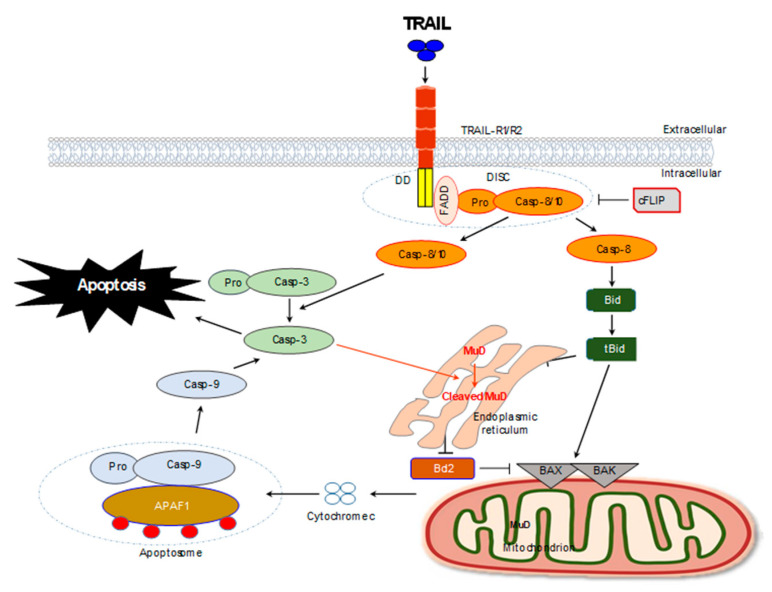
TRAIL-mediated apoptotic signaling pathway overview: Binding of TRAIL to cognate TRAIL receptor, TRAIL-R1/2 results in receptor oligomerization and recruitment of Fas-associated protein with death domain (FADD) and caspase-8 to form a functional death-inducing signaling complex (DISC). Upon DISC formation, caspase-8/-10 is cleaved and activated, which in turn can cleave and activate caspase-3 and the BH3-only protein Bid. Active, cleaved BID (tBid) can bind to pro-apoptotic BAX and BAK, resulting in mitochondrial membrane permeabilization and release of mitochondrial proteins cytochrome c. Cytochrome c, apoptotic protease-activating factor 1 (APAF1) and caspase-9 combine with ATP to form a functional apoptosome that results in cleavage and activation of caspase-9, which can then cleave caspase-3. Caspase-3 can cleave a large number of intracellular targets resulting in the morphological and biochemical hallmarks of apoptosis (Stuckey and Shah, 2013 [63]). Upon TRAIL stimulation, MuD activation by caspase-3 occurs prior to Bcl-2 cleavage and Bid may be essential for the regulation of MuD expression and function upon TRAIL stimulation. Abbreviations: TRAIL, tumor necrosis factor (TNF)-related apoptosis-inducing ligand; APAF-1, apoptotic protease activating factor-1; Bid; BH3-interacting domain death agonist; Bcl-2, B cell chronic lymphocytic leukemia/lymphoma 2; BAK, Bcl-2 homologous antagonist/killer; BAX, Bcl-2-associated protein; FADD, Fas-associated death domain; cFLIP, cellular FLICE-inhibitory protein; DISC, death-inducing signaling complex.

**Table 1 ijms-21-05583-t001:** Established applications of TRAIL anti-cancer therapy.

TRAIL Application	Modification	Deliverables	References
Addition of a leucine zipper	LZ-TRAIL	Facilitates TRAIL stability of trimeric formation in vitro and in vivo leading to enhanced anti-cancer efficiency	[19]
Radioiodinated recombinant human TRAIL	(^125^I) rhTRAIL	Detection of in vivo biodistribution and clearance of cancer cells	[46]
Addition of FLAG tag to TRAIL	FLAG-TRAIL	Enhances antitumor efficiency via FLAG tag that allows crosslinking of TRAIL by using an anti-FLAG antibody	[50]
Fusion of the extracellular domain of Flt3L and an isoleucine zipper to the N terminus of TRAIL. GFP fused to C terminus	S-TRAIL-GFP	Enhanced apoptosis via bystander effect and stabilized oligomerization of TRAIL. GFP expression allows visualization of TRAIL expressing cells-enhanced antitumor activity and detection	[49]
Addition of an isoleucine zipper	iz-TRAIL	Facilitated oligomerization resulting in improved cytotoxicity-enhanced antitumor activity	[50]
scFv–sTRAIL bifunctional fusion proteins	scFv-sTRAIL	Enhanced antitumor activity by increasing specificity and strength of TRAIL response. Permits paracrine signaling	[52,57]
Human serum albumin [23] genetically fused to N-terminus of secretable TRAIL	HSA-Flag-TNC-TRAIL	Efficacy increased through increased serum half-life to improve bioavailability	[53]
PEGylated TRAIL attached to transferrin	Transferrin-PEG-TRAIL (Tf-PEG10K-TRAIL)	Combined enhancement in tumor targeting/killing properties	[54]
PEGylation of TRAIL with PEGa of different molecular weights	PEG-TRAIL	Increased serum half-life and protracted activity in vivo leading to enhanced efficacy and longevity	[55]
Nanocomplex system	TRAIL-loaded PLGAb microspheres	Efficacy and longevity increased via sustained TRAIL release and tumor killing properties in vivo	[56]
Luciferase fused to the N terminus of sTRAIL	SRL0L2TR	Direct extracellular visualization and monitoring of levels, time of delivery, and localization of stem cell-delivered proteins by bioluminescent imaging enabling detection of tumor cells	[58]
Multifunctional nanoparticle comprising doxorubicin (DOX) and pORF–hTRAIL	pORF-hTRAIL	Anti-glioma efficacy in vivo increased	[59]
Ultrasound contrast agents chemically conjugated to TRAIL	TRAIL-UCA	Detection enhanced for ultrasound imaging	[60]
TRAIL conjugated to ferric oxide nanoparticles	Nanoparticle-TRAIL	Enhances antitumor activity in glioma and glioma stem cells in vitro and in vivo	[61]
Adenoviral infection of secretable trimeric TRAIL	Human UCB-MSC	Irradiation enhances U87-MG tumor tropism and therapeutic potential of SCs	[62]
Lentiviral infection of secretable TRAIL	Human BM-MSCs	Use of real-time imaging to follow migration and therapeutic effect of MSCs on primary and established human GBM tumor	[63]
Non-viral nucleofection of TRAIL	Human A-MSCs	Reduction of tumor volume and significant survival benefit in vivo in rat glioma models	[64]
Adenoviral transduction of dodecameric TRAIL	Rat BM-MSCs	Complete elimination of established renal cell carcinoma (RCC) in vivo	[65]
Secretable TRAIL	Human MSC/mouse MSC	Stem cells are eliminated after therapeutic effect by addition of the prodrug gancyclovir established GBMs	[66]
Secretable TRAIL introduced using nonviral PEI(600)-Cyd	MSCs	Reduction in lung metastasis	[67]
Lentiviral infection of secretable TRAIL	Mouse NSCs	PI-103 augments in vivo response of GBM6/8/12 in vitro, Gli36-EGFRVIII in vivo gliomas to TRAIL	[68]
Lentiviral infection of secretable TRAIL	Mouse NSCs	Synergism with TRAIL resulting in eradication of Human U87-MG established glioma model tumor in vivo locked nucleic acid (LNA) anti-miR-21 oligonucleotides	[69]
LV-TRAIL under tet promoter	Human MSCs	Cleared metastatic disease in lung through conditional expression of TRAIL using DOX	[70]
Secretable TRAIL	Human A-MSCs	Decrease in malignant fibrous histiocytoma metastasis	[71]
Secretable TRAIL	Mouse NSC/human MSC	Stem cells encapsulated in sECM. Increased retention of stem cells within established and primary GBMs	[72]
Inducible TRAIL	Human MSC	Halts breast cancer growth and decreases degree of bone and lung metastasis via stem cells encapsulated in silk scaffold	[73]
LV-EGFR-nanobody TRAIL (ENb2-TRAIL)	Mouse NSC	Targets EGFR and TRAIL signaling pathways simultaneously on GBMs	[74]

Abbreviations: PEG, polyethylene glycol; PLGA, poly(lactic-co-glycolic acid); UCB-MSCs, umbilical cord blood-derived mesenchymal stem cells; BM-MSCs, bone marrow-derived mesenchymal stem cells; A-MSCs, adipose-derived mesenchymal stem cells; NSCs, neural stem cells; NPCs, neural progenitor cells; sECM, synthetic extracellular matrix; EGFR, epidermal growth factor receptor.

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
