# Peer review of "The MUDENG Augmentation: A Genesis in Anti-Cancer Therapy?"

_ijms, 2020, doi:10.3390/ijms21155583_

Round 1

Reviewer 1 Report

This review is well written and interesting. Approach to the subject is innovative and  provide interesting review by the latest reports and data. It summarize previous achievements and applications regarding TRAIL in anticancer therapy but also reference to the novel protein MuD. The review is enriched with many figures and tables which add significant value to this paper. 

However some minor corrections must be implemented. In the first lines of introduction (29-36) there are banal and obvious sentences. In my opinion its inadequate and unappealing introduction to such comprehensive review.

In this review authors have mentioned about TRAIL resistance in cancer cells and some ways to overcome this problem. It would be a great advantage for this paper to expand this issue and include more references and examples, which compounds can sensitize cancer cells to TRAIL. Because its not only chemiotherapeutics but also some naturally occuring compounds (like polyphenols).

In the line 192 please correct the spelling. Not "pro-apopotactic and anti-apopotactic" but pro-apoptotic and anti-apoptotic.

In the lines 254-262 there is a serious mistake. Fas (CD95/APO-1) is a receptor indeed. However TRAIL (CD253, TNFSF10, APO-2) is not. It's a ligand which binds to TRAIL receptors DR4 and DR5 (and also few others). So we can use a phrase TRAIL receptors, which means a group of receptors, but we cannot say that TRAIL (CD253, TNFSF10, APO-2) belongs to the death receptor family, because it' s not a receptor. We can say it belongs to TNF superfamily (which contains ligands and receptors). This paragraph must be certainly improved.

Author Response

This review is well written and interesting. Approach to the subject is innovative and provide interesting review by the latest reports and data. It summarize previous achievements and applications regarding TRAIL in anticancer therapy but also reference to the novel protein MuD. The review is enriched with many figures and tables which add significant value to this paper.

Ans: We would like to thank the Editor and the reviewers for their time and efforts towards our manuscript. Thankyou for the revision opportunity. We also appreciate your valuable suggestions and comments towards upgrading the quality of the manuscript. We have now revised the manuscript according to your comments. Thank you.

However some minor corrections must be implemented. In the first lines of introduction (29-36) there are banal and obvious sentences. In my opinion its inadequate and unappealing introduction to such comprehensive review.

Ans: We apologize; we have now revised the introduction. Thank you.

In this review authors have mentioned about TRAIL resistance in cancer cells and some ways to overcome this problem. It would be a great advantage for this paper to expand this issue and include more references and examples, which compounds can sensitize cancer cells to TRAIL. Because it is not only chemiotherapeutics but also some naturally occuring compounds (like polyphenols).

Ans: We have added a short discussion on this in the revision and have cited publications that have actually exclusively concentrated and reported on this. Thank you.

In the line 192 please correct the spelling. Not "pro-apopotactic and anti-apopotactic" but pro-apoptotic and anti-apoptotic.

Ans: Sorry corrected. Thank you.

In the lines 254-262 there is a serious mistake. Fas (CD95/APO-1) is a receptor indeed. However TRAIL (CD253, TNFSF10, APO-2) is not. It's a ligand which binds to TRAIL receptors DR4 and DR5 (and also few others). So we can use a phrase TRAIL receptors, which means a group of receptors, but we cannot say that TRAIL (CD253, TNFSF10, APO-2) belongs to the death receptor family, because it' s not a receptor. We can say it belongs to TNF superfamily (which contains ligands and receptors). This paragraph must be certainly improved.  

Ans: Yes. It is our mistake. We corrected the sentence as you pointed out. Line258-262: “ Fas (CD95/APO-1) and TRAIL (CD253, TNFSF10, APO2) are members of a subset of the TNF receptor superfamily known as ‘death receptors’ [66]. To date, the overwhelming majority of studies on Fas and TRAIL (TNF-related apoptosis-inducing ligand) have explored the role of these receptors as initiators of apoptosis. It is reported that although transformed cells frequently express Fas and TRAIL, most do not undergo apoptosis interacting with these receptors.” → this has been revised as below:

 “Fas (CD95/APO-1) and TRAIL (CD253, TNFSF10, APO2) receptor are members of a subset of the TNF receptor superfamily known as ‘death receptors’ [66]. To date, the overwhelming majority of studies on Fas and TRAIL-R have explored the role of these receptors as initiators of apoptosis. It is reported that although transformed cells frequently express Fas and TRAIL-R, most do not undergo apoptosis interacting with these cognate ligands.”

Thank you for rightly pointing out.

Thankyou for your constructive suggestions.

Reviewer 2 Report

In this review, the authors sum-up the current knowledge on both TRAIL-R agonist-based anti-cancer therapy and the potential of MUDENG targeting. This topic is an interesting one. Below are several comments to improve the manuscript.

In general, the English language is sometimes not precise and the grammar needs to be improved throughout the manuscript. Maybe this could be done with the help of a native English-speaking colleague. I have highlighted some examples below, but this needs to be thoroughly checked for the whole manuscript.

The first two sentences of the abstract are extremely general and not really needed. Same comment for the first paragraph (lines 29-36) of the introduction which can be shortened or removed.

Paragraph from line 65-71 is a bit repetitive and can be shortened.

« With nanotechnology having perpetuated every form of science and development and innovative medicine, why would it spare cancer alone»: perpetuated is not the right term here.

Line 81: risks not “chances”.

The English language in the last paragraph of the introduction (lines 84-89) needs to be refined. For example: « This review addresses the highlights on the inputs from TRAIL towards cancer therapy » would read better like this: this review first describes the potential of using TRAIL-R agonists for cancer therapy. Also, the authors should refrain from stating “this is the first review that shows” throughout their manuscript.

Line 98: “TRAIL system is one such highly evolved therapeutic approach»: this sentence needs rephrasing as the TRAIL system is not a therapeutic approach.

Line 106: “This has paved the way for establishing the way to an entirely novel dimension of cancer research and therapy [35] »: repetitive.

Line 122: “always” not needed.

Line 128: “launched” is not the appropriate term: “have been developed” might be better.

Line 132: improved, not “improvised”.

Line 136: there is a problem with bibliographic tool used for one of the references.

Line 138: “too” is not needed.

Table 1: LZ-TRAIL “trimerization activity”: activity is not the right term here.

z-TRAIL: correct nomenclature is iz-TRAIL

HSA-TRAIL: TRAIL missing in first column

PEGylated TRAIL: “leading to efficiency” is in italic

Inducible TRAIL: “breast and bone metastasis model” repetitive with preceding sentence.

Line 162-163: This resistance can occur from binding to cleavage of the effector caspases, along any point of the TRAIL-signaling pathway. This sentence is difficult to understand and needs rephrasing.

Line 182: « evolutionary conserved » : throughout the animal kingdom ? just eukaryotes ?.. Please specify.

Can the authors comment on the possibility that MUDENG might impact on endocytosis or trafficking of TRAIL-Rs? Was this ever checked for in the literature?

Are there any reported mutations or SNPs for MUDENG (and if so, what are the consequences)?

Line 194: Typo: “Case”

Line 195: what does “near” mean here?

Line 196: “plausible reasons” is incorrect: the authors need to specify how they reach this conclusion. I guess the authors meant Bid cleavage in the same sentence?

Line 216: overexpressed instead of overexposed.

Line 220: restrain from strating sentences with And. There is a typo for signaling and mediated.

Line 223: interact, not “meet”. The whole paragraph needs re-writing. Line 230: Do the author suggest CRIPSR-based KO or siRNA in cancer cells as a therapy here? What other strategy could be proposed instead?

Line 233: this sentence is not needed: “This is the first review which reviews the achievements of MuD ».

« It is reported that although transformed cells frequently express Fas and TRAIL, most do not undergo apoptosis interacting with these receptors. » Do the author talk about the ligands or the receptors here?

Line 264 : « Their report somehow brought TRAIL and MuD onto a common platform. » I do not understand this sentence.

Line 268 : « same as » should be replaced by concomitantly to

Please rephrase this « Depletion of MuD reduced TRAIL stimulation in cell viability. »

How do the authors conceptually explain the anti- and pro-apoptotic effects of MuD? Do the author think that the cleaved forms of MuD possess any activity towards cell death, was this actually evaluated in the literature?

« Graviola is established for its anti-cancer effect. » Please add a reference for this statement.

Ref 55 should be cited along refs 18 and 19 on line 60.

Figures 1 and 2 are extremely general and do not add any meaningful message as compared to other published reviews on anti-cancer therapies in general or agonists of TRAIL-Rs. I would suggest to remove these and instead prepare one figure depicting the different known domains (and when relevant, known interactors and corresponding functions), caspase cleavage sites, etc… of MuD. Figure 3: the different levels at which MuD impacts the signalling should be depeicted.

It would be more readable and look more homogenous to choose only one classically-used font (like Helevetica, Times, Arial) and solely use this throughout the different figures.

I find part 4 very poorly written. I do not think statements like ‘to a world seeking solutions for cancer, it is strange that there are not many takers.’ is adding anything to this review and strongly advise for removing this type of opinion statement.

Author Response

In this review, the authors sum-up the current knowledge on both TRAIL-R agonist-based anti-cancer therapy and the potential of MUDENG targeting. This topic is an interesting one. Below are several comments to improve the manuscript.

Ans: We would like to thank the Editor and the reviewers for their time and efforts towards our manuscript. Thankyou for the revision opportunity. We also appreciate your valuable suggestions and comments towards upgrading the quality of the manuscript. We have now revised the manuscript according to your comments. Thank you.

In general, the English language is sometimes not precise and the grammar needs to be improved throughout the manuscript. Maybe this could be done with the help of a native English-speaking colleague. I have highlighted some examples below, but this needs to be thoroughly checked for the whole manuscript.

Ans: We sincerely apologize for this. We have now had the manuscript thoroughly reviewed by a native English speaker. The Manuscript will now be rid of all language issues. Very kind of you to go through sentence by sentence and suggest modifications. We have carried out all your suggestions. Thank you for your patience. 

The first two sentences of the abstract are extremely general and not really needed. Same comment for the first paragraph (lines 29-36) of the introduction which can be shortened or removed.

Ans: Yes we have followed your suggestion in the revision.

Paragraph from line 65-71 is a bit repetitive and can be shortened.

Ans: Yes shortened.

« With nanotechnology having perpetuated every form of science and development and innovative medicine, why would it spare cancer alone»: perpetuated is not the right term here.

Ans: True, changed

Line 81: risks not “chances”.

Ans: Changed.

The English language in the last paragraph of the introduction (lines 84-89) needs to be refined. For example: « This review addresses the highlights on the inputs from TRAIL towards cancer therapy » would read better like this: this review first describes the potential of using TRAIL-R agonists for cancer therapy. Also, the authors should refrain from stating “this is the first review that shows” throughout their manuscript.

Ans: Sure, we have revised this section. Thank you.

Line 98: “TRAIL system is one such highly evolved therapeutic approach»: this sentence needs rephrasing as the TRAIL system is not a therapeutic approach.

Ans: rephrased.

Line 106: “This has paved the way for establishing the way to an entirely novel dimension of cancer research and therapy [35] »: repetitive.

Ans: deleted.

Line 122: “always” not needed.

Ans: deleted

Line 128: “launched” is not the appropriate term: “have been developed” might be better.

Ans: corrected

Line 132: improved, not “improvised”.

Ans:corrected

Line 136: there is a problem with bibliographic tool used for one of the references.

Ans: Yes. As mentioned, we corrected the wrong references.

Line 138: “too” is not needed.

Ans: deleted.

Table 1: LZ-TRAIL “trimerization activity”: activity is not the right term here.

Ans: Yes. We changed sentences as follows: Facilitates trimerization activity and stability in vitro -- → Facilitates TRAIL stability of trimeric formation in vitro --

z-TRAIL: correct nomenclature is iz-TRAIL

Ans: corrected

HSA-TRAIL: TRAIL missing in first column

Ans: added

PEGylated TRAIL: “leading to efficiency” is in italic

Ans: corrected

Inducible TRAIL: “breast and bone metastasis model” repetitive with preceding sentence.

Ans: removed.

Line 162-163: This resistance can occur from binding to cleavage of the effector caspases, along any point of the TRAIL-signaling pathway. This sentence is difficult to understand and needs rephrasing.

Ans: rephrased

Line 182: « evolutionary conserved » : throughout the animal kingdom ? just eukaryotes ?.. Please specify.

Ans: Line181-182 → new Line 167: “—We revised this sentence as follows: “-- This gene has been evolutionary conserved--” → “This gene has been evolutionarily conserved in some mammals, suggesting —"

We confine the meaning of this sentence to mammals Based on the information obtained from Reference [80]. Lee et al. tracked the genetic sequencing of MuD in various mammals, including cow, dog, mouse, and rat, and showed 85-92% high sequence identity compared with human MuD gene sequence.

Reference [80]: Lee, M. R et al. A novel protein, MUDENG, induces cell death in cytotoxic T cells. BBRC 2008, 370, (3), 504-8.

Can the authors comment on the possibility that MUDENG might impact on endocytosis or trafficking of TRAIL-Rs? Was this ever checked for in the literature?

Ans: It is interesting question. Current research on MuD function can be explained in two ways. One is involved in the Endocytosis process. The second is in the TRAIL-mediated apoptosis. Which is right, is still unanswered, because MuD may have both functions. Anyway, we are focusing our research on MuD's (in) direct relevance on TRAIL-mediated apoptotic signaling rather than relevance on endocytosis and trafficking of TRAIL-Rs, thus, we would like to hold the answer at the fact that MuD's function has connectivity between endocytosis and apoptosis.

Are there any reported mutations or SNPs for MUDENG (and if so, what are the consequences)?

Ans: The literature concerned is yet to be published

Line 194: Typo: “Case”

Ans: corrected

Line 195: what does “near” mean here?

Ans: revised

Line 196: “plausible reasons” is incorrect: the authors need to specify how they reach this conclusion. I guess the authors meant Bid cleavage in the same sentence?

Ans: Yes. We corrected these sentences:

“MuD resides mainly at the ER and minorly mitochondria regulates apoptotic signaling induced by the formation of TRAIL/TRAIL-R complex. Bid molecules involved in apoptosis and caspase-3 may be essential for the regulation of MuD expression and function. In addition, the formation of 25-kDa Bcl-2 fragments by TRAIL stimulation which converts Bcl-2 from its anti-apoptotic form to the truncated pro-apoptotic form is inhibited by MuD, affecting regulation of caspase activation which subsequently leads to apoptosis (Oncogenesis, 5, e221, doi:10.1038/oncsis2016.30)”.

Line 216: overexpressed instead of overexposed.

Ans: corrected

Line 220: restrain from strating sentences with And. There is a typo for signaling and mediated.

Ans: corrected

Line 223: interact, not “meet”. The whole paragraph needs re-writing.

Ans: Yes. We modified the sentence.

TRAIL and MuD does not interact directly, but MuD gets a signal during TRAIL-mediated apoptotic signaling.”

Line 230: Do the author suggest CRIPSR-based KO or siRNA in cancer cells as a therapy here? What other strategy could be proposed instead?

Ans: Yes. We propose the possibilities using CRIPSR-based KO or siRNA in cancer cells as a therapy. In addition, in line 347-350, we suggest another possibility using TurboID system. If we could find partner molecule(s) that work in collaboration with MuD in apoptotic signaling using TurboID system, MuD, along with this partner molecule(s), could be the center of the target for cancer treatment as well as new biomarker, we have added this information in the future perspective. Thank you.

Line 233: this sentence is not needed: “This is the first review which reviews the achievements of MuD ».

Ans: deleted.

« It is reported that although transformed cells frequently express Fas and TRAIL, most do not undergo apoptosis interacting with these receptors. » Do the author talk about the ligands or the receptors here?

Ans:  Yes. It is our mistake. We corrected the sentence as you pointed out. Line261-262: → It is reported that although transformed cells frequently express Fas and TRAIL-R, most do not undergo apoptosis interacting with these cognate ligands.”

Line 264 : « Their report somehow brought TRAIL and MuD onto a common platform. » I do not understand this sentence.

Ans: Rewritten

Line 268 : « same as » should be replaced by concomitantly to

Ans: changed

Please rephrase this « Depletion of MuD reduced TRAIL stimulation in cell viability. »

Ans: rephrased

How do the authors conceptually explain the anti- and pro-apoptotic effects of MuD? Do the author think that the cleaved forms of MuD possess any activity towards cell death, was this actually evaluated in the literature?

Ans: It is very important question. In 2008, Lee et al for the first time reported that Ectopic expression of MUDENG induced cell death in Jurkat T cells and HeLa cells (BBRC, 370: 504-508, 2008). Subsequently, in the same lab Shin et al showed that MuD is cleaved by active caspase-3 during TRAIL-induced cell death and that this results in a reduction of MuD-induced cytotoxicity (BBRC, 435:234-238,2013). Interestingly our group found that MuD possesses an anti-apoptotic function following TRAIL treatment in astroglioma cells. Both overexpression and knockdown studies show that MuD may enable resistance against TRAIL death stimuli (Oncogenesis, 5, e221, doi:10.1038/oncsis.2016.30).

« Graviola is established for its anti-cancer effect. » Please add a reference for this statement.

Ans: Added

Ref 55 should be cited along refs 18 and 19 on line 60.

Ans: Yes done

Figures 1 and 2 are extremely general and do not add any meaningful message as compared to other published reviews on anti-cancer therapies in general or agonists of TRAIL-Rs. I would suggest to remove these and instead prepare one figure depicting the different known domains (and when relevant, known interactors and corresponding functions), caspase cleavage sites, etc… of MuD.

Ans: We have now clubbed Fig 1 and 2 into a single figure as Revised Figure 1. We felt that the overview could give a birds eye view of the therapies and TRAIL mediated solutions at a glance for the readers, that’s why we want to retain it. We have also put in the known information regarding MuD in Figure 2, with the limited knowledge and resources on MuD there is nothing much to plot. Thank you for your understanding.

Figure 3: the different levels at which MuD impacts the signalling should be depicted. 

Ans: Yes. Thank you for your important point that didn’t strike me before. I have corrected the cartoon. I marked the modified parts in red. We have added the MuD impacts

It would be more readable and look more homogenous to choose only one classically-used font (like Helevetica, Times, Arial) and solely use this throughout the different figures.

Ans: we have used homogenous fonts in figures now. Thank you.

I find part 4 very poorly written. I do not think statements like ‘to a world seeking solutions for cancer, it is strange that there are not many takers.’ is adding anything to this review and strongly advise for removing this type of opinion statement.

Ans: We have revised Section 4.

Do the author think that the cleaved forms of MuD possess any activity towards cell death, was this actually evaluated in the literature?

Ans: Yes. As mentioned above, Shin et al showed that MuD is cleaved by active caspase-3 during TRAIL-induced cell death and that this cleaved MuD reduce MuD-induced cytotoxicity (BBRC, 435:234-238,2013). This paper is the only one published so far regarding the activity of cleaved MuD.

Thank you for your valuable thought provoking comments. We appreciate your efforts and interest. Thank you again.

Round 2

Reviewer 2 Report

The text is clearly imporved as compared to the first version. However, somehow the figures are not included in the pdf, so I cannot check these.

Author Response

The text is clearly imporved as compared to the first version. However, somehow the figures are not included in the pdf, so I cannot check these.

We thank you for your encouraging report. We had attached the figures as a separate file. We are so sorry you could not find them . We have now put them on the manuscript. Thank you for your patience. 

Best

Judy

Round 3

Reviewer 2 Report

The manuscript and figures are nicely improved and I do not have any further comments.